# Clinicopathological Features and Survival Outcomes of Primary Pulmonary Invasive Mucinous Adenocarcinoma

**DOI:** 10.3390/cancers13164103

**Published:** 2021-08-15

**Authors:** Chien-Hung Gow, Min-Shu Hsieh, Yi-Nan Liu, Yi-Hsuan Lee, Jin-Yuan Shih

**Affiliations:** 1Department of Internal Medicine, Far Eastern Memorial Hospital, New Taipei City 220216, Taiwan; gowchien@gmail.com; 2Department of Internal Medicine, National Taiwan University Hospital and College of Medicine, National Taiwan University, Taipei 100225, Taiwan; benson1032@gmail.com; 3Department of Healthcare Information and Management, Ming-Chuan University, Taoyuan 333321, Taiwan; 4Department of Pathology, National Taiwan University Hospital, Taipei 100225, Taiwan; mshsieh065@gmail.com (M.-S.H.); yihsuan65@gmail.com (Y.-H.L.); 5Graduate Institute of Clinical Medicine, National Taiwan University, Taipei 100225, Taiwan; 6Department of Medical Research, National Taiwan University Hospital, Taipei 100225, Taiwan

**Keywords:** invasive mucinous adenocarcinoma, tumor grading, lung, recurrence-free survival, overall survival

## Abstract

**Simple Summary:**

Pulmonary invasive mucinous adenocarcinoma (IMA) is a recognized variant of lung adenocarcinoma (ADC) that has unique histological patterns. Comprehensively clinical studies and pathological analyses on IMAs have been limited because IMA is rarely diagnosed compared with other subtypes. We compared the clinical characteristics, pathological features, and survival outcomes of 77 IMA patients with 520 non-IMA-type ADC patients. Currently, IMAs lack a simple pathological prognostic grading system to predict survival. We therefore proposed a simple two-tier grading system, which was modified from the low- and high-grade PanIN grading system, to evaluate its prognostic value. We found that IMAs have more distinct clinicopathological characteristics compared to non-IMA-type ADCs. For patients with stage I–IIIA IMA, a new two-tier grading system might be useful in predicting recurrence-free survival. We demonstrated that stage I and II IMAs have better overall survival compared with non-IMA-type ADCs.

**Abstract:**

Pulmonary invasive mucinous adenocarcinoma (IMA) has unique histological patterns. This study aimed to comprehensively evaluate the clinicopathological features, prognosis, and survival outcomes of IMAs. We retrospectively identified 77 patients with pulmonary IMA and reviewed their clinical and pathological features. Another 520 patients with non-IMA-type ADC were retrieved for comparison with patients with IMA. A new two-tier grading system (high-grade and low-grade IMAs) modified from the pancreatic intraepithelial neoplasia classification system was used for survival analyses. Compared to patients with non-IMA-type ADC, patients with IMA tended to have never smoked (*p* = 0.01) and had early-stage IMA at initial diagnosis (*p* < 0.001). For stage I–II diseases, the five-year overall survival (OS) rates were 76% in IMAs and 50% in non-IMA-type ADCs, and a longer OS was observed in patients with IMA (*p* = 0.002). *KRAS* mutations were the most commonly detected driver mutations, which occurred in 12 of the 28 (43%) patients. High-grade IMAs were associated with a shorter recurrence-free survival (RFS) for stage I–IIIA diseases (*p* = 0.010) than low-grade IMAs but not for OS. In conclusion, patients with stage I and II IMA had better OS than those with non-IMA-type ADC. A new two-tier grading system might be useful for predicting RFS in stage I–IIIA IMAs.

## 1. Introduction

Lung adenocarcinoma (ADC) is the most common primary lung cancer, and its prevalence has increased in recent years. Pulmonary invasive mucinous ADC (IMA), a distinct histological variant of lung ADC, accounts for approximately 5% of lung ADCs [1]. IMAs have a distinct invasive capacity, a predominant architectural pattern, and an amount of mucin production, which mostly originate from the bronchial epithelium or submucosal glands [2]. Previous studies have suggested that IMAs might have more distinguished clinical, genetic, and pathological features compared to conventional lung ADCs [3,4,5].

A comparison of clinical characteristics between IMAs and other subgroups of lung ADCs has been reported in previous studies. A lack of significant differences in demographic information, such as age, gender, race, and smoking status, was observed between IMAs and non-mucinous ADCs [5,6]. However, patients with IMA tended to have an earlier tumor stage than those with other types of lung ADC [6]. Another study demonstrated that in stage IV cases, patients with IMA frequently had lower lobe tumor mass, initial bilateral lung involvement, and pneumonia-like consolidation on chest tomography [5,7].

Driver mutation analyses of IMAs showed a unique pattern. *KRAS* mutation is the most common driver mutation in IMAs, with a prevalence of 28–87% of cases [4,8,9,10,11]. Other common driver mutations discovered in IMAs were neuregulin 1 (*NRG1*) fusion, including *CD74-NRG1*, *SLC3A2-NRG1*, and *VAMP2-NRG1*. All NRG1 fusion proteins showed their oncogenic functions in ERBB2/ERBB3 signaling [12,13,14]. Other mutations frequently detected in conventional ADCs, such as *EGFR* mutations, *BRAF* mutations, *ALK* fusion, and *ROS1* fusion, were rarely detected in IMAs [4].

Histologically, typical IMAs are low-grade ADCs comprising tall columnar cells with abundant intracellular or extracellular mucins that form mixed lepidic and invasive acinar patterns [1]. Some IMAs can have high-grade features with micropapillary patterns, prominent nucleoli, frequent mitoses, decreased mucin production, and tumor necrosis. Morphologically, IMAs share similar pathological features with pancreatic intraepithelial neoplasia (PanIN) since low-grade IMAs resemble low-grade PanINs (including PanIN1 and PanIN2), whereas high-grade IMAs resemble high-grade PanINs (PanIN3). Several pathological features have been evaluated and possibly associated with prognostic values, such as mucin spread size [15], invasive size [16], spread through air spaces (STAS) [17], proportion of goblet cells [18], expression of thyroid transcription factor-1 [19], and predominant growth patterns [20]. However, these parameters alone do not consider the relative heterogeneity of IMAs. In 2020, a study group from United Kingdom (UK) proposed a two-tier grading system for IMAs based on scoring of 5 parameters including differentiation, nuclear atypia, mitosis, necrosis, and lymphovascular invasion [21]. Most recently, a grading system based on predominant and high-grade patterns has been proposed by the International Association for the Study of Lung Cancer Pathology Committee (IASLC) as a predictive marker for survival in invasive pulmonary ADCs [22]. Despite this, the IASLC grading system is recommended in the latest WHO classification; it is used for resected early-stage invasive non-mucinous lung adenocarcinoma and not for IMA [23]. Currently, IMAs still lack a simple prognostic grading system to predict survival.

To date, comprehensive driver mutation analyses and clinical studies on IMAs have been limited because IMA is rarely diagnosed compared with other subtypes. Additionally, the heterogeneous characteristics of IMA tumors and patients led to conflicting results regarding prognosis. The present study aimed to define the clinical features, pathological patterns, and molecular characteristics of IMAs. We proposed a simple two-tier grading system, which was modified from the low- and high-grade PanIN grading system, to evaluate its prognostic value. We also compared the clinicopathological features and survival outcomes of IMA with non-IMA-type ADC patients.

## 2. Materials and Methods

### 2.1. Study Design

We conducted a retrospective study assessing the consecutive pathological records of pulmonary IMAs at two medical centers in Taiwan, the National Taiwan University Hospital and the Far Eastern Memorial Hospital, from January 2011 to December 2017. A total of 90 IMA cases were identified. All histological specimens were retrieved to confirm the diagnosis. Thirteen cases were excluded because of a disqualified diagnosis according to the IMA criteria. Finally, 77 IMA cases were included in this study. We also retrieved the clinical data of 520 ADC patients diagnosed with non-IMA-type ADC based on surgical and biopsy evaluations in the same period for comparison with patients with IMA. We recorded the clinicopathological characteristics of the 597 patients, including age, gender, smoking status, initial stage at diagnosis [24], Eastern Cooperative Oncology Group (ECOG) performance status score [25], metastatic sites, history of other primary malignancies, and treatment. Recurrence-free survival (RFS) was defined as the time from surgical resection to disease relapse. Overall survival (OS) was defined as the period from the date of the initial diagnosis of lung cancer to the date of death (or censored at the date of last follow-up or loss of contact). The study was approved by the Institutional Review Board of the National Taiwan University Hospital and Far Eastern Memorial Hospital.

### 2.2. Mutation Analyses

Molecular analyses of the driver mutations were performed. Specimen preparation and DNA/RNA extraction were performed as previously described [26]. Reverse transcription polymerase chain reaction and sequencing of *EGFR*, *KRAS*, *HER2*, *BRAF*, and *MET* exon 14 skipping mutations, along with *ALK*, *RET*, and *ROS1* fusion analyses, were performed individually as described previously [26]. Immunohistochemistry staining was applied in a portion of *ALK* fusion analyses, and fluorescence in situ hybridization was performed for *ROS1* fusion examinations as previously described [26]. *CD74-NRG1* fusion was determined as described previously [27]. The primers used for *SLC3A2-NRG1* and *VAMP2-NRG1* fusions and *HER2* transmembrane domain (TMD) mutation are described in the Appendix A.

### 2.3. Histologic Grade

We proposed a new two-tier IMA grading system based on the current PanIN classification system [28] to evaluate the prognostic value of IMA. The reasons for using the PanIN grading system are as follows. First, IMA cells share morphological similarities with PanIN. Second, a well-defined two-tier pathological grading system for PanIN has already been established as PanIN1/PanIN2 for low-grade IMAs, whereas PanIN3 is a high-grade lesion with a higher risk of progression to pancreatic ductal ADC. Low-grade IMA (IMA-G1) was defined as tumor cells with pathological changes similar to low-grade PanIN, including PanIN1-like cells (flat tall columnar mucinous epithelial cells with basally located small round nuclei, inconspicuous nucleoli, and rare mitoses) and/or PanIN2-like cells (mixed flat and papillary mucinous epithelial cells with mild nuclear abnormalities, including nuclear crowding, pseudostratification, small nucleoli, and occasional mitoses). High-grade IMA (IMA-G2) was defined as tumor cells exhibiting cytological and architectural changes similar to high-grade PanIN (PanIN3), with enlarged nuclei, prominent nucleoli, decreased mucin production, presence of mitoses and/or necrosis, and arranged micropapillary or filigreed patterns. Tumor cells with cribriform, solid, or sarcomatoid patterns were also classified as high-grade IMAs. In addition to the grading, the presence of satellite nodules away from the main tumor (skipping lesions) or STAS was also recorded. STAS was defined as previously described [17]. The available slides of each case were thoroughly reviewed by two experienced thoracic pathologists (M.-S.H. and Y.-H.L.) using the two-tier grading system.

In addition, we also used the IMA grading system proposed by the UK group who used 5 parameters, including differentiation, nuclear atypia, mitosis, necrosis, and lymphovascular invasion, to give a total score that ranged from 2 to 9 [21]. The differentiation is based on the predominant histologic pattern of IMA as well differentiated (lepidic, score 1), moderately differentiated (acinar/papillary, score 2), or poorly differentiated (solid/micropapillary/cribriform, score 3). Nuclear atypia is based on nuclear pleomorphism and presence of distinct nucleoli. It is categorized as mild atypia (relatively uniform nuclei with indistinct nucleoli at 100 magnification, score 1), moderate atypia (relatively uniform nuclei with distinct nucleoli at 100 magnification, score 2), or severe atypia (bizarre, enlarged nuclei of varied sizes, with some nuclei at least twice as large as others, score 3). Mitotic activity is recorded as <4 per 2 mm^2^ (score 0) or ≥4 per 2 mm^2^ (score 1). Tumor necrosis is absent (score 0) or present (score 1). Lymphovascular invasion is absent (score 0) or present (score 1). IMA is considered as low-grade when the total score is between 2–5 and high-grade when the total score is between 6–9 [21].

### 2.4. Statistical Analyses

All categorical variables of clinicopathological characteristics were analyzed using Pearson’s chi-squared test. Univariate and multivariate analyses for RFS were performed using Cox regression model. Survival curve analyses for RFS and OS were estimated using the Kaplan–Meier method and compared using the log-rank test. Two-sided *p*-values less than 0.05 were considered statistically significant. All statistical analyses were performed using International Business Machines (IBM) Statistical Package for the Social Sciences Statistics for Windows version 26.0 (IBM Corp., Armonk, NY, USA).

## 3. Results

### 3.1. Clinical Characteristics of Patients with Invasive Mucinous Adenocarcinoma (IMA) and Non-IMA-Type Adenocarcinoma

The clinical characteristics of the 77 patients with IMA and 520 patients with non-IMA-type ADC are listed in Table 1. The non-IMA-type ADC cases included 256 *EGFR* mutations (106 exon 19 deletions, 117 exon 21 L858R, 15 exon 20 insertions, and 18 rare mutations in exons 18 and 21), 29 *ALK* fusions, 22 *BRAF* V600E mutations, 20 *KRAS* mutations, 20 *HER2* mutations, 13 *MET* exon 14 deletion (*MET*∆14) mutations, 9 *ROS1* fusions, and 151 undetected mutations in the driver genes. For patients with IMA, the median age was 62.2 (35–90) years. Moreover, 69% (53 of 77) were women, 16% (12 of 77) were current/former smokers, and 9% (12 of 77) had a history of other primary malignancies. Furthermore, 88% of the patients had early-stage IMA (stage I, 69%; stage II, 9%; and stage IIIA, 10%) at initial diagnosis. No stage IIIB patient was retrieved in our IMA group. There were no significant differences in age, sex, ECOG performance score, and history of other primary malignancies between patients with IMA and patients with non-IMA-type ADC. Patients with IMA tended to have never smoked (*p* = 0.01) and had early-stage (I-IIIA) at initial diagnosis (*p* < 0.001).

### 3.2. Mutation Analyses of IMAs

We examined multiple major driver mutations in patients with IMA. Only a portion of IMA cases were tested because of inadequate or non-available tumors in our laboratory. Detailed mutational data are listed in Table 2. Among them, *KRAS* mutations were the most commonly detected driver gene alteration in IMAs, which occurred in 12 of 28 (43%) cases. Other driver mutations were rarely detected, including a case with *HER2* V659E in TMD mutation (1 of 14, 7%), a case with *CD74-NRG1* fusion (1 of 17, 6%), two cases with *ALK* fusion (2 of 34, 6%), a case with *ROS1* fusion (1 of 22, 5%), and two cases with *EGFR* mutations (2 of 47, 4%). Other driver alterations, such as *HER2* kinase domain mutation, *BRAF* V600E mutation, *RET* fusion, *MET*∆14 mutation, *SLC3A2-NRG1* fusion, and *VAMP2-NRG1* fusion, were not detected.

### 3.3. Recurrence-Free Survival and Overall Survival in Patients with Stage I–IIIA IMA

We next determined whether clinicopathological characteristics would contribute to survival outcomes in 68 patients with stage I–IIIA IMA. Among clinical features, we analyzed several factors, including age, gender, smoking status, concurrent primary malignancies, and driver mutations. For RFS analysis, none of the above clinical factors showed significant differences in patients with IMA. Available tissues in pathologic analyses were 57 for satellite nodules/skipping lesions, 57 for STAS, 61 for tumor necrosis, and 61 for IMA-2 grade evaluations. In univariate analyses, significantly poor RFS features were observed in tumors with STAS (hazard ratio (HR): 5.28, 95% confidence interval (CI): 1.02–27.22, *p* = 0.047) and in tumors with high-grade two-tier IMA grading system (two groups) (HR: 2.12, 95% CI: 1.14–3.94, *p* = 0.018) (Table 3). Subsequently, we included all pathological features for multivariate analysis. Only a high-grade two-tier IMA grading system (HR: 2.66; 95% CI: 1.30–5.47, *p* = 0.008) was an independent prognostic factor and was significantly associated with a shorter RFS. For OS analysis, we observed that only patients who had history of other primary malignancies were significantly associated with a shorter OS (HR: 6.34, 95% CI: 1.28–31.5, *p* = 0.024). Therefore, OS for multivariate analysis was not assessed. Clinical tumor size (T1/T2 vs. T3/T4) was not included in the analysis because patients with T1/T2 tumors were well known and strongly associated with longer RFS (*p* < 0.001) and OS (*p* = 0.017).

We next compared the results of our two-tier IMA grading system (based on the presence of absence of morphologically PanIN3-like areas) and the grading system proposed by UK group using five parameters. These two systems gave the same results in 60 of 61 IMA cases. The only case with inconsistent result had PanIN3-like areas but low nu-clear atypia. Using UK grade for survival analyses, both RFS (HR: 1.78, 95% CI: 0.95–3.37, *p* = 0.073) and OS (HR: 0.88, 95% CI: 0.30–2.57, *p* = 0.811) failed to demonstrate a prognostic value.

### 3.4. Clinical Application of Two-Tier IMA Grading System for Survival Outcomes in Patients with Stage I-IIIA IMA

Representative images of low-grade IMA (IMA-G1) and high-grade IMA (IMA-G2) in pulmonary IMA are shown in Figure 1. The clinical characteristics of low- and high-grade IMA in patients with stage I–IIIA pulmonary IMA are listed in Table 4. Among these characteristics, there were no significant differences in age, smoking status, ECOG status score, history of other primary malignancies, and driver mutation status between low- and high-grade IMA. High-grade IMA (IMA-G2) tumors were more significantly observed in female patients with IMA compared to low-grade cases, but the difference was not statistically significant (*p* = 0.063). Only clinical late stage was associated with high-grade IMA (IMA-G2) (stage I/II/III, 54%/15%/30% in IMA-G2 vs. 85%/8%/6% in IMA-G1; *p* = 0.02).

We next evaluated RFS and OS according to the two-tier IMA grading system in patients with IMA. We found that in patients with stage I–IIIA IMA, a shorter RFS was observed in patients with high-grade IMA (IMA-G2) tumors (*p* = 0.010) but not in OS (*p* = 0.480; Figure 2a). Similar results were noted when only examined in patients with stage I and II IMA (RFS, *p* = 0.042; OS, *p* = 0.710; Figure 2b). However, when we only examined stage I IMA for RFS, events were nonremarkable (four in low- and two in high-grade IMA); therefore, the result showed no statistical significance (*p* = 0.196).

### 3.5. Overall Survival in Different Stages of Patients with IMA Compared to Patients with Non-IMA-Type ADC

We first analyzed OS in early (I-IIIA) or advanced (IIIB-IV) stages of IMA and non-IMA-type ADC. Patients with stage I–IIIA IMA had longer OS than patients with non-IMA-type ADC (*p* = 0.001; Figure 3a); however, there was no statistically significant difference in median OS in stage IIIB-IV between two groups (*p* = 0.856; Figure 3b). We next analyzed OS at different stages. Because of the limited number of stage II IMA (*n =* 7) and non-IMA-type ADC (*n =* 29), we combined stage I and II in both groups for OS analysis. Although the median OS was not reached in both groups, patients with stage I–II IMA had longer OS than patients with non-IMA-type ADC (*p* = 0.002) (Figure 3c). Similar results were observed in stage I cases (*p* = 0.007; Appendix A). The five-year OS rates in patients with stage I–II diseases were 76% in IMAs and 50% in non-IMA-type ADCs. For stage IIIA disease, there was no statistically significant difference in median OS between IMA and non-IMA-type ADC (*p* = 0.891; Figure 3d). The OS for stage IIIB was not analyzed because there was no stage IIIB patient in our IMA group. Finally, the median OS was 12.5 months (95% CI: 0.0–25.9 months) in patients with stage IV IMA and 14.6 months (95% CI: 11.1–18.1 months) in patients with stage IV non-IMA-type ADC. There was no statistically significant difference in OS in stage IV between groups (*p* = 0.972; Appendix A).

## 4. Discussion

Pulmonary IMA have been reported to represent a clinically and pathological unique subtype of non-small cell lung cancer (NSCLC) [3,4,5]. In the present study, we supported this finding in a large cohort of patients with IMA and compared it with non-IMA-type ADC. We also proposed and applied a new two-tier IMA grading system similar to the pancreatic intraepithelial neoplasm grading system (PanIN) to evaluate whether this grading system could improve the prediction of recurrence of the disease in patients with IMA. This grading system is simple and gives almost the same result as another more complex scoring system, which needs to score five pathological parameters [21]. Furthermore, we observed that the earlier stage (stage I–II) of IMA had better OS than that in non-IMA-type ADC, whereas both groups had similar OS in late-stage (stage III-IV) disease. These results provide an additional vision for clinical practice and prognostic prediction of pulmonary IMA.

In this study, we are unable to distinguish patients with IMA with non-IMA-type ADC based on general clinical features, such as age, gender, smoking status, ECOG score, and history of other primary malignancies in Taiwanese, an Eastern Asian population. Although IMAs are thought to be diagnosed at an advanced inoperable stage, we observed that they tended to be found in the early stage; 69% of our cases had stage I disease. Lee et al. have reported that half of the patients with resected IMA in a Korean cohort had stage I IMA (51%) [29]. In another study, Moon et al. have shown a similar result, that is, approximately 70% of IMAs were either stage I or stage II at initial diagnosis [7]. For non-IMA-type ADC, our results showed that a significant portion of tumors were diagnosed at an advanced inoperable stage (52% with stages IIIB-IV disease), which was different from that in patients with IMA. The stage distribution for patients with non-IMA-type ADC is close to that reported in a national cancer database survey for NSCLC [30]. Therefore, we consolidated the significant difference in stage distribution at initial diagnosis between IMA and non-IMA-type ADC. The reason why IMA cases are more often diagnosed at an early stage remains unclear; it is possible that IMA is associated with a lower rate of nodal metastases and less lymphatic invasion [8,31]. Although it might be possible to diagnose some IMA cases as non-IMA-type ADC on small biopsies, we believe that the majority of IMA cases can have an accurate diagnosis on transthoracic core needle biopsy based on cellular features [5,31].

Although comprehensive mutational analyses in IMA tumors are limited because of their rarity in lung ADC, molecular characteristics are significantly different from those observed in non-IMA-type ADC. Our data showed predominant mutations in *KRAS* codon 12 glycine (43%) and rare *EGFR* mutations (4%), which is consistent with the results of published studies in Asian populations [5,32]. Previous studies have demonstrated that *KRAS* mutations can be detected in approximately 70% of IMAs that carry different types of substituted mutations at glycine 12 in Western populations [3,6,33] and had concurrent mutations in 35% of *KRAS* mutation-positive IMAs, including *STK11*, *GNAS*, *TP53*, *CDKN2A*, and *PIK3CA* [6]. We suggest that the discrepancy in *KRAS* mutation rates between studies might be associated with racial differences. Given the high prevalence of *KRAS* G12 mutations in our cohort as well as previous reports, it would be interesting to further evaluate the potential role of new *KRAS* G12C-inhibitors (such as sotorasib or adagrasib) in IMA patients harboring *KRAS* G12 mutations [34,35]. *ALK* fusion and *ROS1* fusion were previously considered to be significantly rare gene alterations in IMAs [5]. However, we detected two *ALK* fusions (6%) and one *ROS1* fusion (5%) in this study. Similar findings for *ALK* fusion cases have been reported in previous studies [3,6,33]. Nevertheless, *ALK* fusion-positive IMAs should be carefully reviewed again by pathologists since *ALK* fusion-positive ADC with extensive mucin production can mimic IMA patterns [36]. Although *ROS1* fusion was rarely found in IMA patients, previous cases have been reported in early-stage IMAs [37,38]. Another gene alteration, *NRG1* fusion, occurring in 8–32% of IMA tumors, was reported at a low prevalence (<1%) in non-IMA-type ADC [39]. In this study, we only found a case harboring *CD74-NRG1* fusion, but no *SLC3A2-NRG1* or *VAMP2-NRG1* fusion was detected. Finally, a recent study demonstrated that *HER2* TMD V659E mutations occurred in 0.14% of lung ADCs [40], and another study demonstrated that 5 of 14 (35%) cases with *HER2* TMD were IMAs [41]. We only detected that 1 of 14 (7%) IMA tumors harbored the *HER2* TMD V659E mutation. It would be interesting to further evaluate whether *HER2* TMD V659E mutations are more frequently observed in IMAs and to determine the potential treatment efficacy of afatinib in such patients.

Because IMA shares similar pathological features with PanIN, we proposed a two-tier grading system for IMA. IMA is categorized as low-grade (IMA-G1) when it shows low-grade PanIN1 and PanIN2 features. If there is any area showing high-grade PanIN (PanIN3)-like features, the tumor is classified as high-grade IMA (IMA-G2). In our case series, it was not difficult to detect the high-grade, PanIN3-like component because the high-grade area typically comprised more than 10% of the total tumor and sometimes was the predominant component of the tumor. Using this two-tier grading system, we found that high-grade IMA (IMA-G2) was associated with poor RFS in stage I–II or I–IIIA pulmonary invasive ADCs. We observed that low-grade IMA (IMA-G1) tended to have stage I disease, while high-grade IMA (IMA-G2) tumors tended to have stage IIIA disease. Therefore, this two-tier grading system could provide a potential prognostic grouping for RFS in IMAs. In this study, our grading system (based on the absence/presence of high-grade PanIN-like areas) gives almost the same result as that from the five-parameter scoring system proposed by UK group [21]. As low-grade and high-grade PanIN patterns are well recognized by general pathologists, our grading system is much simpler to apply.

Recently, a new grading system was proposed by IASLC to predict the survival prognosis for invasive pulmonary ADC [22]. This IASLC grading system comprises a combination of predominant and any tumor with 20% or more high-grade patterns and was designed for the heterogeneity of invasive pulmonary ADCs. According to the IASLC system, invasive ADC can be classified as grade 1 (well-differentiated) with lepidic predominant with no or less than 20% of high-grade patterns, grade 2 (moderately differentiated) with acinar or papillary predominant with no or less than 20% of high-grade patterns, and grade 3 (poorly differentiated) with any tumor with 20% or more of high-grade patterns. They reported that higher grade was associated with reduced RFS and OS in either stage I–III or stage I cases only [22]. However, this IASLC grading system is recommended to be used in invasive non-mucinous adenocarcinoma, and there is still no established grading system for IMA recommended by WHO [23]. Regardless of their grading, IMA cells usually grow along the alveolar septa with a superficial spreading pattern or skip satellite nodules with a so-called “aerogenous” spreading pattern along with extracellular mucin produced by IMA. Their unique growing and spreading patterns make IMAs distinct from conventional pulmonary invasive ADCs. Based on the observation that IMA shares similar cytological and architectural changes with PanIN, we used a simple, two-tier grading system instead of three grades in the IASLC grading system. Nevertheless, there are still similarities between these two grading systems. Low-grade IMA with PanIN1 and PanIN2 features correlates with IASLC lepidic/acinar/papillary patterns with grade 1 and grade 2 nuclear features, whereas high-grade IMA with PanIN3 features correlates with IASLC micropapillary/solid/complex gland patterns and grade 3 nuclear features. Other histologic features, such as STAS, have been reported to have prognostic value in predicting recurrence [17], and our data showed a consistent result that IMA with STAS had shorter RFS. However, the prognostic value was less significant than that of the two-tier IMA grading system in multivariate analysis (Table 3). For OS analysis, the two-tier IMA grading system failed to demonstrate a prognostic value in OS because the majority of the patients with stage I and II IMA, and very few patients with stage IIIA IMA, survived more than 10 years, and further follow-up duration is needed to determine whether the two-tier IMA grading system has a role in predicting OS. In this study, we found 2 of 9 patients with stage I IMA carrying secondary cancers (one had oral cancer and the other had hepatocellular carcinoma) died earlier due to progression of the second tumor, which led to stage I-IIIA IMAs with second malignancies had shorter OS.

In this study, we observed a stage-dependent OS between the IMA and non-IMA-type ADC. The five-year OS rates in stage I patients were 78% in IMAs and 65% in non-IMA-type ADCs. Due to the small number of patients with stage II IMA, we categorized stages I and II together and observed that patients with IMA also had better OS than patients with non-IMA-type ADC. The five-year OS rates in patients with stage I–II IMA were 76% in IMAs and 50% in non-IMA-type ADC. Although previous lectures suggested an aggressive clinical course for mucinous BAC (or IMA) [29,42], our data were similar to those of other studies that defined IMA as low-to intermediate-grade lung ADC [29,43]. Certainly, we observed that 85% of stage I IMA tumors were low-grade IMAs. However, more aggressive behavior was noted in stage IIIA tumors, with four of seven (57%) classified as high-grade IMA. The OS observed in stage IIIA IMAs was not significantly different from that in patients with non-IMA-type ADC. Similar findings were found in patients with stage IV IMA and non-IMA-type ADC. There were no differences in OS between the two groups, consistent with the result of a previous study [5].

The present study has several potential limitations. First, it is a retrospective analysis study, the sample size of IMAs is relatively small, and molecular analyses are not available for all IMA tumors. Second, this study had an extreme imbalance in patient numbers between the two compared groups (IMA and non-IMA type ADC), which is intrinsic to the rarity of the subtype but limits the affordability of the analyses. Third, several censored patients with stage I and II IMAs are still alive at the last follow-up date; we suggest that a prolonged observation may be needed for future work. Furthermore, we only included patients with samples obtained based on surgical or computerized/ultrasonography-guided biopsy evaluations in both groups. Some ADCs diagnosed from aspiration or effusion block were not included in this study. Nevertheless, this study provides important insights into the unique clinicopathological features, prognostic evaluation, and survival outcomes of patients with IMA. Further large-scale prospective studies are necessary to clarify overcome these limitations.

## 5. Conclusions

IMAs have more distinct clinicopathological characteristics compared to non-IMA-type ADCs. Patients with IMA tend to have never smoked, be early-stage IMA at initial diagnosis, and have predominant *KRAS* codon 12 mutations. For patients with stage I–IIIA IMA, a new two-tier grading system might be useful in predicting RFS. Patients who had history of other primary malignancies had poor OS. Finally, our study demonstrated that stage I and II IMAs have better OS compared with non-IMA-type ADCs, whereas no survival benefit was noted between the two groups in patients with stage III and IV IMAs.

## Figures and Tables

**Figure 1 cancers-13-04103-f001:**
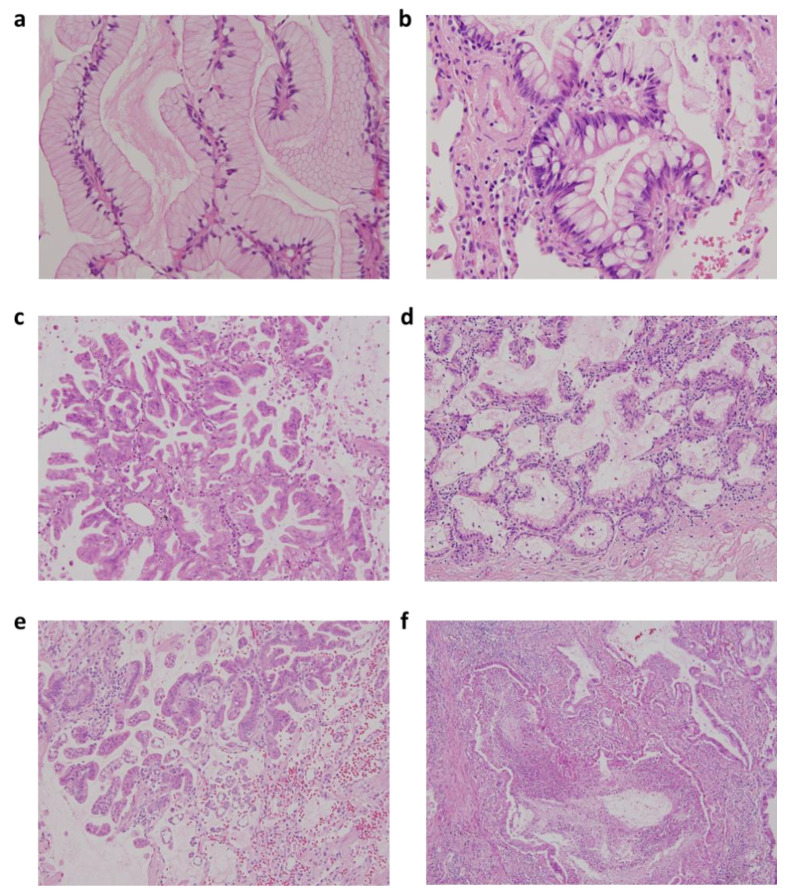
Representative figures of low-grade and high-grade invasive mucinous adenocarcinoma (IMA) according to two-tier IMA-2 grading system in IMA. (**a**) Low-grade IMA with pancreatic intraepithelial neoplasia 1 (PanIN1)-like features with flat mucinous epithelium and basally located nuclei; (**b**) low-grade IMA with PanIN2-like features with nuclear crowding, pseudostratification, and tuft-like growth; (**c**) high-grade IMA with PanIN3-like features with micropapillary or filigreed patterns, decreased mucin production, and vesicular nuclei with conspicuous nucleoli; (**d**) low-grade IMA with mixed PanIN1 and PanIN2-like areas; (**e**) high-grade IMA with mixed PanIN2 and PanIN3-like areas along with STAS; and (**f**) high-grade IMA with complex glands and tumor necrosis (original magnification: A and B, 400×; C–F, 200×).

**Figure 2 cancers-13-04103-f002:**
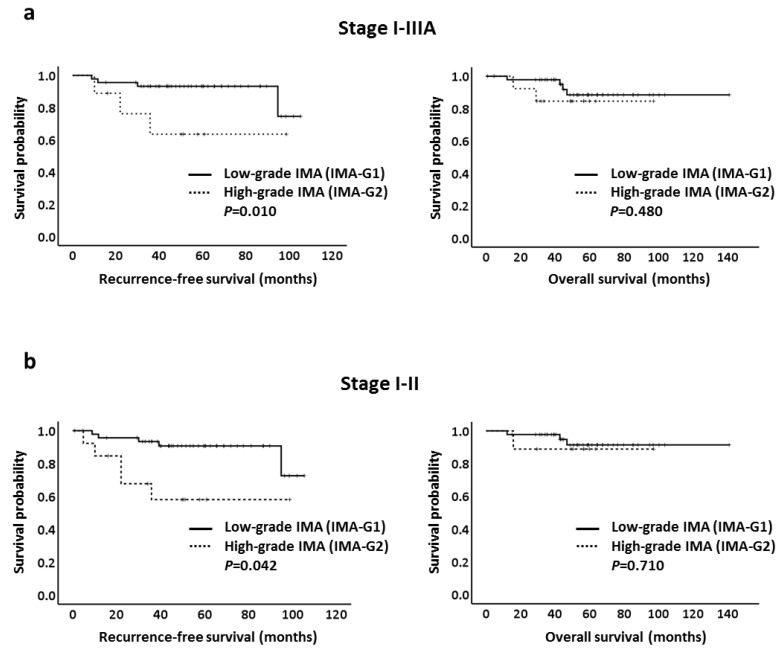
Kaplan–Meier curves of recurrence-free survival and overall survival in pulmonary invasive mucinous adenocarcinoma patients with low-grade invasive mucinous adenocarcinoma (IMA) (IMA-G1) and high-grade IMA (IMA-G2) according to two-tier IMA-2 grading system. (**a**) Stage I–IIIA; (**b**) stage I–II.

**Figure 3 cancers-13-04103-f003:**
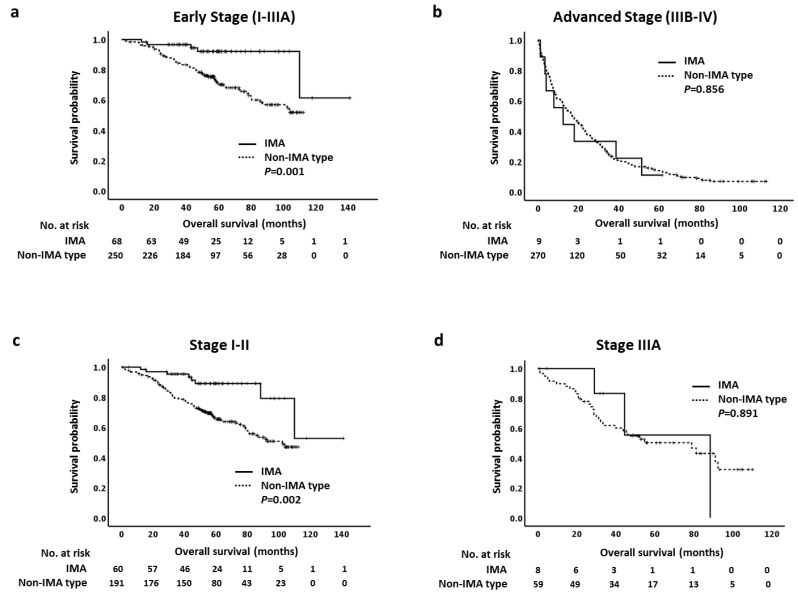
Kaplan–Meier curves of overall survival of different stages in pulmonary invasive mucinous adenocarcinoma (IMA) and non-IMA-type adenocarcinoma. (**a**) Early stage (I–IIIA), (**b**) advanced stage (IIIB-IV), and (**c**) stage I-II, (**d**) stage IIIA.

**Table 1 cancers-13-04103-t001:** Clinical characteristics of patients with pulmonary invasive mucinous adenocarcinoma (IMA) (*n* = 77) or non-IMA-type adenocarcinoma (*n =* 520).

Clinical Characteristic	IMA	Non-IMA-Type ADC	*p*-Value ^#^
Patients, *n*	77	520	
Age, years			0.087
Median (range)	62.2 (35–90)	64.4 (32–90)	
≥65, *n* (%)	29 (38)	252 (49)	
Sex, *n* (%)			0.063
M	24 (31)	223 (43)	
F	53 (69)	297 (57)	
Smokers, *n* (%)			0.010 *
Current/ever	12 (16)	159 (31)	
Never	65 (84)	361 (69)	
ECOG PS score, *n* (%)			0.263
0−1	71 (92)	453 (87)	
2−4	6 (8)	67 (13)	
Stage, *n* (%)			<0.001 *
I	53 (69)	162 (31)	
II	7 (9)	29 (6)	
IIIA	8 (10)	59 (11)	
IIIB	0 (0)	30 (6)	
IV	9 (12)	240 (46)	
Early/Advanced stage			<0.001 *
I-IIIA	68 (88)	250 (48)	
IIIB-IV	9 (12)	270 (52)	
History of other primary malignancies, *n* (%)			1.000
Only lung cancer	68 (88)	461 (89)	
Other cancers	9 (12)	59 (11)	

Abbreviations: ECOG, Eastern Cooperative Oncology Group; F, female; M, male; *n*, number; PS, performance status; SD. ^#^
*p*-values were calculated using a two-sided chi-squared test. ***** Values that are statistically significant (*p* < 0.05).

**Table 2 cancers-13-04103-t002:** Driver mutations of available tumor samples in 77 patients with pulmonary invasive mucinous adenocarcinoma.

Driver Mutation	No. of Sample Tested/Not Tested	Mutation Rate, % (No. of Mutation/Tested)	Aminoa Acid Mutation (*n*)
*KRAS* mutation	28/49	43 (12/28)	G12A (2), G12C (3), G12D (6), G12V (1)
*HER2* TMD mutation	14/63	7 (1/14)	V659E (1)
*CD74-NRG1* fusion	17/60	6 (1/17)	
*ALK* fusion	34/43	6 (2/34)	
*ROS1* fusion	22/55	5 (1/22)	
*EGFR* KD mutation	47/30	4 (2/47)	delE746-A750 (1), M825L (1)
*HER2* KD	27/50	0 (0/27)	
*BRAF* V600E	25/52	0 (0/25)	
*RET* fusion	17/60	0 (0/17)	
*MET* exon 14 skipping	17/60	0 (0/17)	
*SLC3A2-NRG1* fusion	18/59	0 (0/18)	
*VAMP2-NRG1* fusion	18/59	0 (0/18)	

Abbreviations: del, deletion; KD, kinase domain; No. (*n*), number; and TMD, transmembrane domain.

**Table 3 cancers-13-04103-t003:** Clinicopathological analyses of recurrence-free survival (RFS) and overall survival (OS) in stage I–IIIA pulmonary invasive mucinous adenocarcinoma.

	Recurrence-Free Survival				Overall Survival	
Factor	Univariate		Multivariate ^#^		Univariate	
	HR (95% CI)	*p*-Value	HR (95% CI)	*p*-Value	HR (95% CI)	*p*-Value
Age, years						0.074
<65 (*n* = 44)	1				1	
≥65 (*n* = 24)	0.63 (0.19–2.01)	0.436			3.71 (0.88–15.62)	
Sex						0.517
Male (*n =* 22)	1				1	
Female (*n =* 46)	0.39 (0.19–1.91)	0.392			0.62 (0.15–2.62)	
Smokers						0.254
Current/ever (*n =* 10)	1				1	
Never (*n =* 58)	0.51 (0.07–3.98)	0.523			2.60 (0.50–12.43)	
History of other primary						0.024 *
malignancies						
Only lung (*n =* 59)	1				1	
Other cancers (*n =* 9)	1.47 (0.32–6.81)	0.622			6.34 (1.28–31.5)	
Driver mutation						0.870
Undetected (*n =* 15)	1				1	
Yes (*n =* 53)	1.69 (0.51–5.63)	0.391			1.15 (0.22–5.95)	
Satellite nodules/ skipping						0.642
lesions						
No (*n =* 27)	1		1		1	
Yes (*n =* 30)	3.08 (0.62–15.26)	0.169	2.62 (0.49–13.7)	0.255	1.53 (0.25–9.18)	
Spread through air spaces						0.422
(STAS)						
No (*n =* 31)	1		1		1	
Yes (*n =* 26)	5.28 (1.02–27.22)	0.047 *	2.12 (0.28–18.9)	0.503	2.09 (0.35–12.53)	
Tumor necrosis						0.624
No (*n =* 55)	1		1		1	
Yes (*n =* 6)	2.19 (0.46–10.43)	0.324	2.27(0.36–14.2)	0.381	1.71 (0.20–14.66)	
IMA-2 grade						0.487
Low (IMA-G1) (*n =* 48)	1		1		1	
High (IMA-G2) (*n =* 13)	2.12 (1.14–3.94)	0.018 *	2.66 (1.30–5.47)	0.008 *	1.35 (0.58–3.16)	

Abbreviations: F, female; M, male; *n*, number; and IMA-2 grade, two-tier invasive mucinous adenocarcinoma grading system. **^#^** Multivariate analysis included all pathological features and was calculated using the backward stepwise method for the Cox regression model. ***** Values that are statistically significant (*p* < 0.05).

**Table 4 cancers-13-04103-t004:** Clinical features of low-grade IMA (IMA-G1) and high-grade IMA (IMA-G2) in stage I–IIIA pulmonary invasive mucinous adenocarcinoma (*n =* 61).

Clinical Characteristic	Low Grade (IMA-G1)	High Grade (IMA-G2)	*p*-Value ^#^
Patients, *n*	48	13	
Age, years			1.000
<65	31 (65)	8 (62)	
≥ 65, *n* (%)	17 (35)	5 (38)	
Sex, *n* (%)			0.063
M	17 (35)	3 (23)	
F	31 (65)	10 (77)	
Smokers, *n* (%)			1.000
Current/ever	7 (15)	1 (8)	
Never	41 (85)	12 (92)	
ECOG PS score, *n* (%)			1.000
0−1	46 (95)	12 (92)	
2−4	2 (5)	1 (8)	
Stage, *n* (%)			0.020 *
I	41 (85)	7 (54)	
II	4 (8)	2 (15)	
IIIA	3 (6)	4 (30)	
History of other primary malignancies, *n* (%)			0.350
Only lung cancer	43 (90)	10 (77)	
Other cancers	5 (10)	3 (23)	
Driver mutations, *n* (%)			0.200
*KRAS*	8 (17)	2 (15)	
Others	1 (2)	2 (15)	
No	39 (81)	9 (70)	

Abbreviations: ECOG, Eastern Cooperative Oncology Group; F, female; M, male; *n*, number; PS, performance status; and SD, standard deviation. ^#^
*p*-values were calculated using a two-sided chi-squared test. ***** Values that are statistically significant (*p* < 0.05).

## Data Availability

Data is contained within the article or Appendix A.

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
