# Peer review of "Clinicopathological Features and Survival Outcomes of Primary Pulmonary Invasive Mucinous Adenocarcinoma"

_cancers, 2021, doi:10.3390/cancers13164103_

Round 1

Reviewer 1 Report

Pulmonary invasive mucinous adenocarcinoma (IMA) has unique histological patterns. This is a retrospective study, the authors aimed to evaluate the clinicopathological features, prognosis, and survival outcomes of IMA. This paper retrospectively identified 77 patients with pulmonary IMA and reviewed their clinical and pathological features. Another 520 patients with non-IMA-type were retrieved for comparison with patients with IMA. Currently, IMAs lack a simple pathological prognostic grading system to predict survival. The authors proposed a simple two-tier grading system, which was modified from the low- and high-grade PanIN grading system, to evaluate its prognostic value. They found IMAs have more distinct clinicopathological characteristics compared to non-IMA-type ADC. For patients with stage I–III IMA, a new two-tier grading system might be useful in predicting recurrence-free survival. The authors demonstrated that stage I and II IMAs have better overall survival compared with non-IMA-type ADC. The study is intriguing and may provide important information about IMA.

Major comments:

  1. The authors emphasized about their new simple two-tier grading system. Please offer the data using the old grading system and compared the differences in these two systems.
  2. Did stage I-III IMA patients all received operation? Or only in stage I-to-IIIa IMA patients, not in stage IIIb patients. If so, please offer or classified patients as stage I-IIIa (early stage, can receive operation) and stage IIIb-IV (advanced stage, cannot receive operation) in Table 1, Table 4, Figure 2, and Figure 3.

Reviewer 2 Report

In this manuscript, the authors analyze the clinical, mutational, and pathological characteristics of pulmonary invasive mucinous adenocarcinoma (IMA), compared to other adenocarcinoma histotypes. They also propose a two-tier grading system for IMA, mutuated from PaIN. 

Although some evidence already exists about this topic, the manuscript has the merit of analyzing a quite wide case series. Furthermore, the pathologic review of all cases to guarantee the correctness of diagnosis is a significant added value. Overall, the manuscript is clear and well written. Results are fairly described and discussed. There are no major ethical or methodological concerns. 

I only suggest some minor revisions, which may improve the quality of the work:

  • The authors should cite the extreme imbalance in patients number between the two compared groups (IMA and non-IMA NSCLC), which is intrinsic in the rarity of the subtype but limits the affordability of the analyses.
  • Did the authors consider also stage IV cases in the survival analysis? If so, recurrence-free survival is not an appropriate endpoint and its use should be avoided. Given the small number of patients with stage IV disease, I suggest to repeat this analysis excluding them (and specifying it in the text). 
  • In Table 2, the authors should add a line to specify the number of cases for which the molecular tests could not be performed. Indeed, the total number of cases in the table heading should be the same of the overall IMA population. Furthermore, the fact that molecular analyses could be performed only in a subgroup of cases should also be cited in the appropriate Results section, not only in the Discussion.
  • In Table 3, the authors should also add the non-significant p-values obtained at the multivariate analysis.
  • As there is an imbalance in advanced stage diseases between IMA and non-IMA NSCLC (with a higher prevalence of stage III NSCLC in the IMA group), the survival analysis should be corrected for this.
  • The case report of the ROS-positive patient is unnecessary for the comprehension of the work and should be removed from Discussion.
  • The sentence on clinical outcome of stage IV patients treated with chemotherapy should also be removed from Discussion. Indeed, the number of cases is too low to draw any conclusion and we cannot say that their treatment response was poor, as no comparisons have been made with the other subgroup.
  • It would be interesting to add some hypotheses about why IMA cases are more often diagnosed at an early stage (more early symptoms? localization of the lesion? other?).
  • It would be interesting to add some considerations about why the patients with second malignancies had shorter OS (did they die for NSCLC or progression of the second tumor?).
  • In the Discussion, the authors could cite the potential role of new KRAS-inhibitors in the treatment of IMA, given the high prevalence of KRAS G12 mutations in this cohort.
  • In the Conclusion, "To have no smoking status" should be replaced by "To be never smoker" or something similar.

Round 2

Reviewer 1 Report

Congratulations for the excellent paper.